# Validity and reliability of velocity and power measures provided by the Vitruve linear position transducer

**Santiago A. Ruiz-Alias**[1,2], **Deniz Şentürk**[3], **Zeki Akyildiz**[4], **Onat Çetin**[5], **Selman Kaya**[5], **Alejandro Pérez-Castilla**[6,7], **Ivan Jukic**[8,9]*

1 Department of Physical Education and Sports, Faculty of Sport Sciences, University of Granada, Granada, Spain, 2 Sport and Health University Research Center (iMUDS), University of Granada, Granada, Spain, 3 İstanbul Gelişim University School of Physical Education and Sports, Istanbul, Turkey, 4 Sports Science Faculty, Department of Coaching Education, Afyon Kocatepe University, Afyonkarahisar, Turkey, 5 Yalova University, Faculty of Sports Science, Yalova, Turkey, 6 Department of Education, Faculty of Education Sciences, University of Almería, Almería, Spain, 7 SPORT Research Group (CTS-1024), Centro de Investigación para el Bienestar y la Inclusión Social (CIBIS) Research Center, University of Almería, Almería, Spain, 8 Sport Performance Research Institute New Zealand (SPRINZ), Auckland University of Technology, Auckland, New Zealand, 9 Department of Health, Sport and Wellbeing, Faculty of Social and Applied Sciences, Abertay University, Dundee, United Kingdom

* ivan.jukic@aut.ac.nz

**Data Availability Statement:** All relevant data are within the manuscript and its Supporting information files.

## Abstract

This study aimed to determine the validity and between-day reliability of the mean velocity (MV), peak velocity (PV), mean power (MP), and peak power (PP) provided by the Vitruve linear position transducer at different submaximal loads in the free-weight and Smith machine back squat using GymAware as a reference point. Fourteen male sports science students (free-weight back squat one-repetition maximum [1RM]: 132.5 ± 28.5 kg, Smith machine back squat 1RM: 163.9 ± 30.4 kg) performed six experimental sessions, twice per week with 72 hours of rest. The first two included the assessment of the 1RM of both exercises. In the four remaining, both linear position transducers were simultaneously used to record MV, PV, MP, PP of each repetition during an incremental load test (i.e., 20, 40, 60, 80, 90% 1RM) with three minutes of rest between sets. Vitruve displayed both fixed and proportional bias for certain relative loads across all variables. Vitruve did not meet the validity criteria for all (MV, PP) or at least two (MP, PV) relative loads (Coefficient of variation [CV] > 10%; Pearson correlation < 0.70; Effect size > 0.60). MV, PV, MP, and PP recorded by Vitruve displayed acceptable reliability (CV < 10%) with superior reliability observed during a Smith Machine compared to free-weight back squat, and for velocity compared to power variables. Considering GymAware as a reference point, Vitruve was not valid for measuring velocity and power outcomes. Acceptable validity was observed only for PV in the Smith machine back squat, while the other variables—regardless of relative loads and exercise modes—were mostly inaccurate. All variables demonstrated acceptable reliability, with greater reliability noted in the Smith machine compared to the free-weight back squat exercise mode.

**Funding:** The author(s) received no specific funding for this work.

**Competing interests:** The authors have declared that no competing interests exist.

## Introduction

Velocity-based approach to resistance training (VBT) has marked a turning point in resistance training monitoring and prescription practices given its utility for internal and external load management [1–3]. Through the monitoring of the barbell velocity, practitioners can adjust training loads for athletes or clients based on their daily readiness, determine the set termination according to how they are coping with fatigue, as well as analyze their performance status and monitor training progress [1].

Different velocity-monitoring systems are available nowadays [4]. To implement VBT in practice, valid (i.e., the degree to which the device measures what it claims to measure) and reliable (i.e., the consistency of the measure under consistent conditions) devices providing barbell velocity data are needed [5]. In this regard, optoelectronic motion capture systems are considered the gold standard for tracking barbell kinematics [5]. However, a time-consuming calibration process, data collection and post-processing, and the high cost make motion capture systems unfeasible and unaffordable for practitioners. Accordingly, several barbell velocity monitoring devices have emerged that address these limitations [4]. For instance, sports scientists and strength and conditioning professionals frequently use linear position or velocity transducers These devices utilize a cable attached to the barbell to determine the vertical displacement and provide the barbell velocity using the inverse dynamics approach (linear position transducer), or they directly measure velocity by capturing electrical signals proportional to the cable's extension velocity (linear velocity transducer) [5].

Given the continuing development of new velocity-monitoring devices and associated software, the examination of their validity and reliability has been a common concern in the scientific community [6], which requires a continuous update. In this regard, the Speed4lilft linear position transducer attracted great interest among practitioners given its affordable price and reported validity and reliability across different relative loads and exercises [4, 7]. An update of this device was later released under the name Vitruve, in which the manufacturer stated that improvements were made for monitoring faster movements, whose validity was a major concern of the previous version when the barbell velocity was over 1.0 m/s [7]. However, to date, the validity and reliability of kinematic variables provided by Vitruve (i.e., the new version of the device) have been scarcely analyzed. Külkamp et al. [8] recently examined the validity of the barbell displacement measurements provided by Vitruve through pre-defined heights established on a Smith machine, revealing an overestimation of barbell displacement by 1–2 cm, on average. Additionally, Kilgallon et al. [9] have explored the reliability of the load-velocity profiles (20, 40, 60, 80, 90% of the one-repetition maximum [1RM]) of strength-trained males in the free-weight back squat, as well as the validity of the 1RM estimations through different regression models using Vitruve. Across the submaximal loads analyzed, the mean propulsive velocity (MPV), mean velocity (MV), and peak velocity (PV) displayed acceptable reliability (coefficient of variation [CV] < 9%), although large 1RM under and overestimations were observed [9]. Based on the aforementioned, different knowledge gaps need to be addressed, such as determining the validity of Vitruve across various exercise modes (e.g., free-weight and Smith machine back squat), evaluating the validity of other metrics (i.e., mean power [MP], peak power [PP]), and determining its level of agreement with other reference linear position transducers.

It should be noted that the validity and reliability of kinematic variables displayed by linear position transducers during free-weight exercises may be affected by the horizontal motion of the lift [10]. For example, an inclined position at the end of the lift would involve a lower cable displacement, which would therefore reduce the MV [10]. Considering the different angle positions that could be present along the set or a training session, practitioners may prefer to

use linear position transducers with Smith machine exercises given the greater reliability of kinematic variables these devices display in this exercise mode [5]. However, the GymAware linear position transducer incorporates a sensor that measures the angle of the cable, which supposedly corrects the cable displacement according to the horizontal motion of the lift using basic trigonometry [11]. Considering the proven validity of GymAware compared to gold-standard 3D motion capture systems [12–14], it would be of interest to examine the concurrent validity of Vitruve with respect to GymAware in different exercise modes (e.g., free-weight and Smith machine back squat).

The validity and reliability analysis of other variables provided by these devices (i.e., MP and PP) is also needed given the potential discrepancies that might exist between sampling rates and how software manipulates the raw data [5]. Likewise, these variables are widely used by practitioners to determine if their athletes are producing more impulse with the same absolute load (i.e., moving the same absolute load faster) [15]. Therefore, this study aimed to determine the concurrent validity and the between-day reliability of the MV, PV, MP, and PP provided by Vitruve at different submaximal loads (20, 40, 60, 80, and 90% 1RM) in the free-weight and Smith machine back squat exercises using GymAware as a reference point. Based on current knowledge about Vitruve [8, 9], it is hypothesized that the MV and MP might be underestimated. However, it is expected that all variables will display an acceptable reliability.

## Materials and methods

All participants were informed about the research purpose and procedures of the study prior to signing a written informed consent form. The study protocol adhered to the tenets of the Declaration of Helsinki and was approved by the University Human Research Ethics Committee (approval number: 2021/125).

### Study design

After the familiarization session, the participants performed a total of six sessions over three consecutive weeks (twice per week) with 72 hours of rest between the sessions. The first two sessions included the assessment of participants' 1RM in the free-weight and Smith machine back squat exercise modes. Vitruve and GymAware linear position transducers were simultaneously used during the four remaining experimental sessions for monitoring MV, PV, MP, and PP of each repetition across sessions. Throughout the sessions, the sequence of exercise modes was alternated in a randomized counterbalanced manner, with half of the participants commencing the first session with either the free-weight or the Smith machine back squat exercise mode. Participants reported to the laboratory at the same time of the day (± 1 hour) and performed all testing sessions under similar environmental conditions (~22°C and ~40% humidity). The research was conducted under the direct supervision of the same researcher (DS) at the Istanbul Gelisim University Research Laboratory from August 5, 2022, to August 30, 2022.

### Participants

Fourteen male sports science students (age: 21.5 ± 1.5 [range: 19–25] years, body height: 1.77 ± 0.05 m, body mass: 73.7 ± 11.6 kg, free-weight back squat 1RM: 132.5 ± 28.5 kg, Smith machine back squat 1RM: 163.9 ± 30.4 kg) participated voluntarily in this study. A post hoc power analysis was performed using G*Power (Version 3.1) with an effect size (ES) of 1.0 (the minimal difference observed for MV) and an alpha level ($\alpha$) of 0.05. The analysis revealed a statistical power of 0.93. All participants were actively performing resistance training and had at least 6 months of experience with resistance training exercises involved in this study.

Specifically, they declared that they did free-weight and Smith machine back squat exercises in their training routines and were accustomed to the 1RM protocols. Participants were recruited through multiple channels including word of mouth, flyers, and online platforms.

### Familiarization session

The participants' body height and body mass were measured at the beginning of the first familiarization session using an electronic column scale with a fitted stadiometer (Seca 202; Seca Ltd., Hamburg, Germany). Thereafter, participants performed a standardized warm-up protocol consisting of five minutes of running on a treadmill at 7 km/h, mobility exercises (i.e., Prisoner Squat, Lunge to Twist, Good-morning, Lateral Lunge), and 10 back squat repetitions with a free-weight barbell (20 kg) and with the unloaded Smith machine barbell (19 kg) (Technogym, Gambettola, Italy) [16]. Participants then performed three repetitions at 20, 40, and 60% of their self-reported 1RM and one repetition at 80 and 90% 1RM in both back squat exercise modes with three minutes of rest between sets and four minutes between exercise modes. During this session, it was ensured that participants could perform all the lifts with the correct technique. Specifically, the participants started back squat exercises by positioning the bar on the upper trapezius and acromion (high-bar position) with the feet fully stretched in a slightly wider than shoulder-width apart position. The participants squatted down continuously until the angle of their knees reached 90 degrees and the top of their thighs was parallel to the floor. Immediately after reaching this position, they returned to the starting position as quickly as possible. Participants were instructed to keep constant downward pressure on the barbell throughout the whole movement, and they were not allowed to jump off the ground. To ensure adherence to technique requirements, the iPhone 11 Promax (Apple Inc. Cupertino, CA, USA) mobile phone was positioned on a 1-meter-high tripod, 2 meters away from the participants sideways. This was done to monitor hip and knee joint angles of all repetitions, which were recorded in slow motion. Repetitions that were not performed with the desired technique were repeated. Finally, participants were familiarized with the instruction to move the barbell up as fast as they can (i.e., lift at maximal intended velocity). Technique requirements and instructions were standardized across the sessions in this study.

### 1RM testing sessions

Each 1RM testing session began with the same warm-up described above. The 1RM protocol consisted of three repetitions with 20%, 40%, and 60% 1RM, 1 repetition with 80% and 90% of the self-reported 1RM followed by 1RM attempts [17]. After each successful 1RM attempt, the load was increased between two and seven kilograms in consultation with participants until no further weight could be added to the barbell or when they were not able to complete a repetition with a desired technique. A rest period of 3 minutes was provided between all submaximal sets and 1RM attempts.

### Experimental sessions

Each session began with the same warm-up described above. Thereafter, three repetitions at 20, 40, and 60% 1RM and one repetition at 80 and 90% 1RM were performed with three minutes of rest between sets. Both back squat exercise modes were performed in the same week with 72 hours of rest. Testing sessions with both exercise modes were repeated in the following week for the between-day reliability analysis.

## Data acquisition

The linear position transducers GymAware (Power Tool, Kinetic Performance Technologies, Canberra, Australia) and Vitruve (Speed4lift, Madrid, Spain) were positioned on the same sides of the barbell. Both devices were placed directly under the barbell and were attached to it through a Velcro strap. Both devices were connected to their respective applications (Vitruve version: 1.11.2; GymAware version: 2.8) which were installed on a smartphone iPhone 8 Plus (Apple Inc. Cupertino, CA, USA) and a tablet (iPad, Apple Inc. Cupertino, CA, USA). The Vitruve device samples data at 100 Hz [18], meanwhile the GymAware adopts a variable sampling rate that it then down-samples to 50 Hz [19, 20]. More specific details of the sampling method adopted by GymAware are explained by Qaisar et al. [21]. Both linear position transducers internally collect and process the data in a similar fashion [11]. Briefly, a retractable cable is fixed to the axis of an electromechanical sensor. The cable extension produces the rotation of the transducer axis, which converts it into pulses and then to linear displacement. Thereafter, the double differentiation of displacement data permits the calculation of acceleration, from which all the variables reported by these devices are calculated [11].

The software of each device automatically displayed the MV, PV, MP, and PP values. For the GymAware device on day 2, only the highest MV repetition was chosen for each relative load (20, 40, 60, 80, and 90% 1RM) and exercise mode (free-weight and Smith machine back squat) to assess the concurrent validity of Vitruve alongside its corresponding repetition [22]. For the between-day reliability analysis, the repetitions with the highest MV recorded by both devices on days 1 and 2 were selected and compared, for each relative load and exercise mode. These same repetitions were used for analyzing the other variables (PV, MP, PP) [22].

## Statistical analysis

The concurrent validity of the Vitruve with respect to the GymAware was determined through the Pearson product-moment correlation ($r$), CV, and Cohen's $d$ ES. The Vitruve linear position transducer was considered valid if the following criteria were met when the upper (CV) or the lower limits ($r$ and ES) of the 95% confidence intervals (95% CI) did not include 1) CV > 10%; 2) $r < 0.70$; and 3) ES > 0.60 [5]. The standard error of the estimate (SEE) was also calculated and reported. The Bland Altman plots were used to determine fixed and proportional bias. A fixed bias was present when the 95% CI of the mean difference did not contain zero. A proportional bias was present when ($r^2 > 0.1$) [23]. The between-day reliability of MV, PV, MP, and PP displayed by Vitruve was assessed by the CV and CV ratios $= \frac{Highest\ CV}{Lowest\ CV}$. Between-day reliability of these variables was considered acceptable when the 95% CI of CV was < 10% [5]. The smallest detectable difference (SDD), interpreted as the smallest measurement change that corresponds to a real difference beyond zero was calculated for each variable by the standard error of the measurement (SEM) [24]: SDD = 1.96 * $\sqrt{2}$ * SEM. To interpret the magnitude of differences between two CVs (e.g., Smith machine vs. free-weight back squat; mean velocity vs. mean power), a criterion for the smallest important ratio was established as higher than 1.15 [4]. Validity and reliability analyses were performed by means of a custom Excel spreadsheet [25]. Alpha was set at 0.05.

## Results

### Concurrent validity

For MV, Vitruve did not meet the validity criteria for any of the relative loads (ES ≥ -1.1) during the free-weight back squat exercise mode (Fig 1). A fixed negative bias was observed for all relative loads (range: -0.02 to -0.05 m/s) (Table 1). A proportional bias was only observed at

## Mean velocity

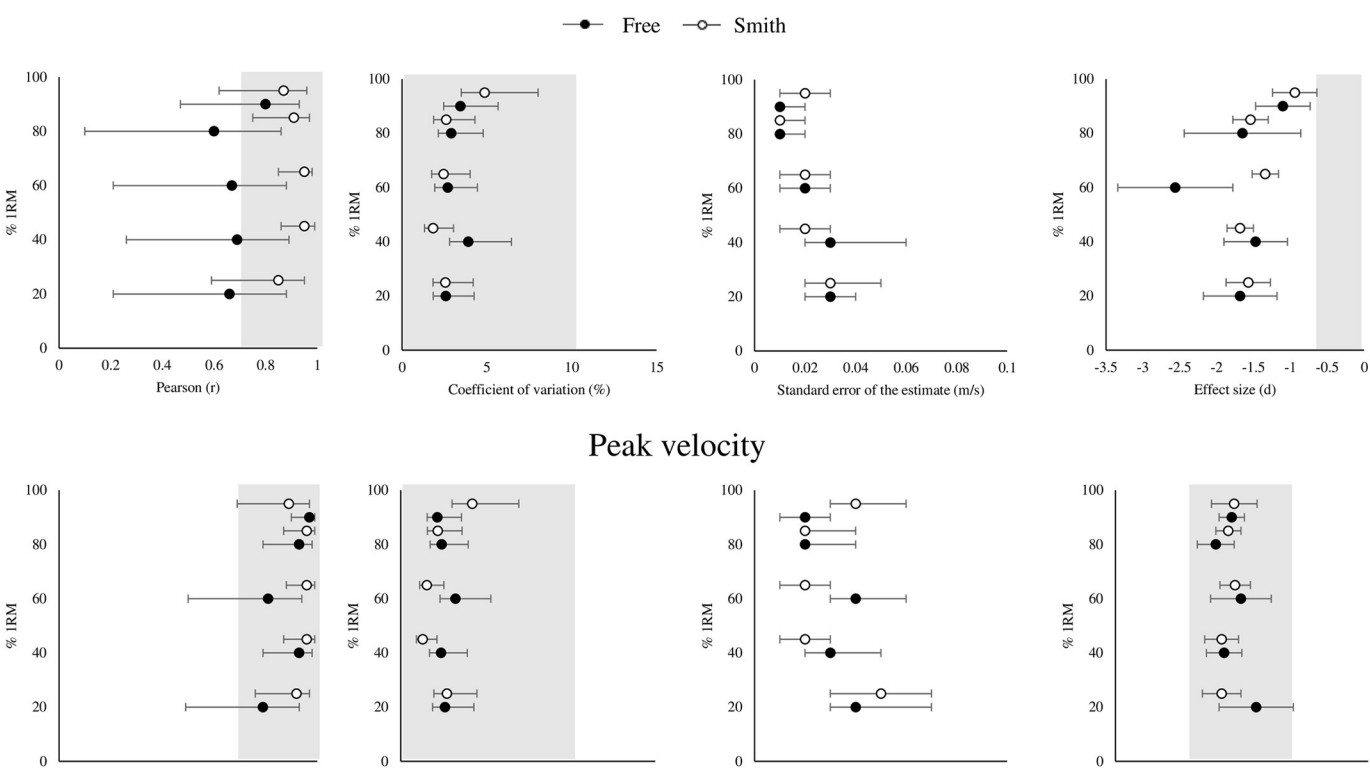

**Fig 1. Validity of Vitruve for the measurement of the mean and peak velocity at different relative loads during the free-weight and Smith machine back squat exercises.** Forest plots displaying Pearson correlation coefficient (r), coefficient of variation (%), standard error of the estimate (m/s) and effect size (d). An acceptable range for validity is shaded in grey.

60% 1RM and 80% 1RM ($R^2 \geq 0.34$). Similarly, during the Smith machine back squat exercise mode, Vitruve did not meet the validity criteria for any of the relative loads (ES $\geq$ -0.94). A fixed negative bias was also observed for all relative loads (range: -0.03 to -0.09 m/s). A proportional bias was only observed at 20% 1RM ($R^2 = 0.16$).

For PV, Vitruve met the validity criteria for 40% 1RM, 80% 1RM, and 90% 1RM ($r \geq 0.93$; CV $\leq$ 2.4%; ES $\leq$ 0.30) but not for 20 and 60% 1RM ($r$ 95% CI < 0.70) during the free-weight back squat exercise mode. No fixed bias was observed across the loads, but a proportional bias was observed at 20, 40 and 60% 1RM ($R^2 \geq 0.12$). During the Smith machine back squat exercise mode, Vitruve met the validity criteria for all relative loads ($r \geq 0.89$; CV $\leq$ 4.2%; ES $\leq$ 0.24). No fixed bias was observed across the loads, but a proportional bias was observed at 40 and 60% 1RM ($R^2 \geq 0.12$).

For MP, Vitruve met the validity criteria for 20% 1RM, 40% 1RM, and 60% 1RM ($r \geq 0.98$; CV $\leq$ 4.5%; ES $\leq$ -0.47) but not for 80 and 90% 1RM (CV 95% CI > 10%) during the free-weight back squat exercise mode (Fig 2). A fixed negative bias was observed for all relative loads (range: -17.5 to -50.8 W). A proportional bias was observed at 20 and 40% 1RM ($R^2 \geq 0.14$). During the Smith machine back squat exercise mode, Vitruve met the validity criteria for 20% 1RM and 40% 1RM ($r \geq 0.98$; CV $\leq$ 4.%; ES $\leq$ -0.50) but not for 60, 80 and 90% 1RM (ES 95% CI > 0.6). A fixed negative bias was observed for all relative loads (range: -33 to -72 W). A proportional bias was observed at 20, 40 and 60% 1RM ($R^2 \geq 0.17$).

**Table 1. Fixed and proportional bias of Vitruve with respect GymAware for the mean velocity, peak velocity, mean power, and peak power at different relative loads during the free-weight and Smith machine back squat exercises.**

| Back squat exercise | Relative load (%1RM) | Mean velocity (m/s) | | | Peak velocity (m/s) | | | Mean power (W) | | | Peak power (W) | | |
|---|---|---|---|---|---|---|---|---|---|---|---|---|---|
| | | Bias (95% CI) | Slope (95% CI) | $R^2$ | Bias (95% CI) | Slope (95% CI) | $R^2$ | Bias (95% CI) | Slope (95% CI) | $R^2$ | Bias (95% CI) | Slope (95% CI) | $R^2$ |
| Free-weight | 20 | -0.05 (-0.07 to -0.04) | 0.11 (-0.45 to 0.67) | 0.01 | 0.01 (-0.02 to 0.04) | 0.25 (-0.17 to 0.67) | 0.12 | -17.5 (-22.3 to -12.7) | -0.06 (-0.15 to 0.03) | 0.14 | 475 (424 to 528) | 0.51 (0.33 to 0.69) | 0.76 |
| | 40 | -0.07 (-0.09 to -0.05) | -0.20 (-0.73 to 0.32) | 0.05 | -0.02 (-0.04 to 0.00) | -0.16 (-0.40 to 0.07) | 0.16 | -41.2 (-53.1 to -29.3) | -0.11 (-0.22 to 0.00) | 0.30 | 522 (450 to 595) | 0.37 (0.17 to 0.57) | 0.58 |
| | 60 | -0.06 (-0.08 to -0.04) | 0.67 (0.16 to 1.18) | 0.41 | 0.00 (-0.02 to 0.02) | -0.16 (-0.40 to 0.07) | 0.13 | -50.8 (-64.6 to -37.0) | -0.03 (-0.17 to 0.10) | 0.02 | 355 (278 to 432) | 0.11 (-0.16 to 0.39) | 0.06 |
| | 80 | -0.03 (-0.04 to -0.01) | 0.65 (0.08 to 1.23) | 0.34 | -0.02 (-0.03 to 0.00) | -0.03 (-0.27 to 0.21) | 0.01 | -36.2 (-58.0 to -14.3) | -0.06 (-0.28 to 0.15) | 0.03 | 96 (42 to 150) | 0.00 (-0.20 to 0.20) | 0.00 |
| | 90 | -0.02 (-0.03 to -0.01) | 0.00 (-0.41 to 0.42) | 0.00 | -0.00 (-0.02 to 0.00) | 0.07 (-0.08 to 0.23) | 0.08 | -31.5 (-49.2 to -13.9) | -0.02 (-0.23 to 0.18) | 0.00 | 54 (-19 to 127) | 0.00 (-0.30 to 0.30) | 0.00 |
| Smith machine | 20 | -0.08 (-0.09 to -0.06) | -0.24 (-0.59 to 0.11) | 0.16 | -0.03 (-0.05 to 0.00) | -0.08 (-0.34 to 0.18) | 0.03 | -32.9 (-42.4 to -23.4) | -0.13 (-0.25 to 0.00) | 0.28 | 492 (424 to 560) | 0.42 (0.18 to 0.66) | 0.55 |
| | 40 | -0.09 (-0.09 to -0.08) | 0.03 (-0.17 to 0.22) | 0.00 | -0.01 (-0.02 to 0.00) | 0.12 (-0.06 to 0.30) | 0.14 | -52.1 (-57.2 to -47.0) | -0.03 (-0.08 to 0.01) | 0.17 | 556 (469 to 642) | 0.40 (0.11 to 0.69) | 0.44 |
| | 60 | -0.07 (-0.08 to -0.06) | -0.06 (-0.26 to 0.13) | 0.04 | -0.00 (-0.02 to 0.01) | 0.10 (-0.07 to 0.28) | 0.12 | -72.0 (-81.5 to -62.4) | -0.08 (-0.16 to 0.00) | 0.30 | 313 (253 to 374) | 0.25 (0.06 to 0.43) | 0.42 |
| | 80 | -0.04 (-0.05 to -0.04) | -0.01 (-0.27 to -0.26) | 0.00 | -0.01 (-0.02 to 0.00) | -0.04 (-0.22 to 0.14) | 0.02 | -68.4 (-80.1 to -56.8) | -0.00 (-0.13 to -0.13) | 0.00 | 120 (74 to 165) | 0.10 (-0.05 to 0.26) | 0.14 |
| | 90% | -0.03 (-0.04 to -0.02) | -0.01 (-0.35 to 0.32) | 0.00 | -0.01 (-0.03 to 0.01) | 0.05 (-0.25 to 0.35) | 0.01 | -55.1 (-72.5 to -37.7) | -0.17 (-0.31 to -0.02) | 0.35 | 96 (35.8 to 157) | 0.04 (-0.23 to 0.31) | 0.01 |

1RM, one-repetition maximum; 95% CI, 95% Confidence interval; $R^2$: Coefficient of determination.

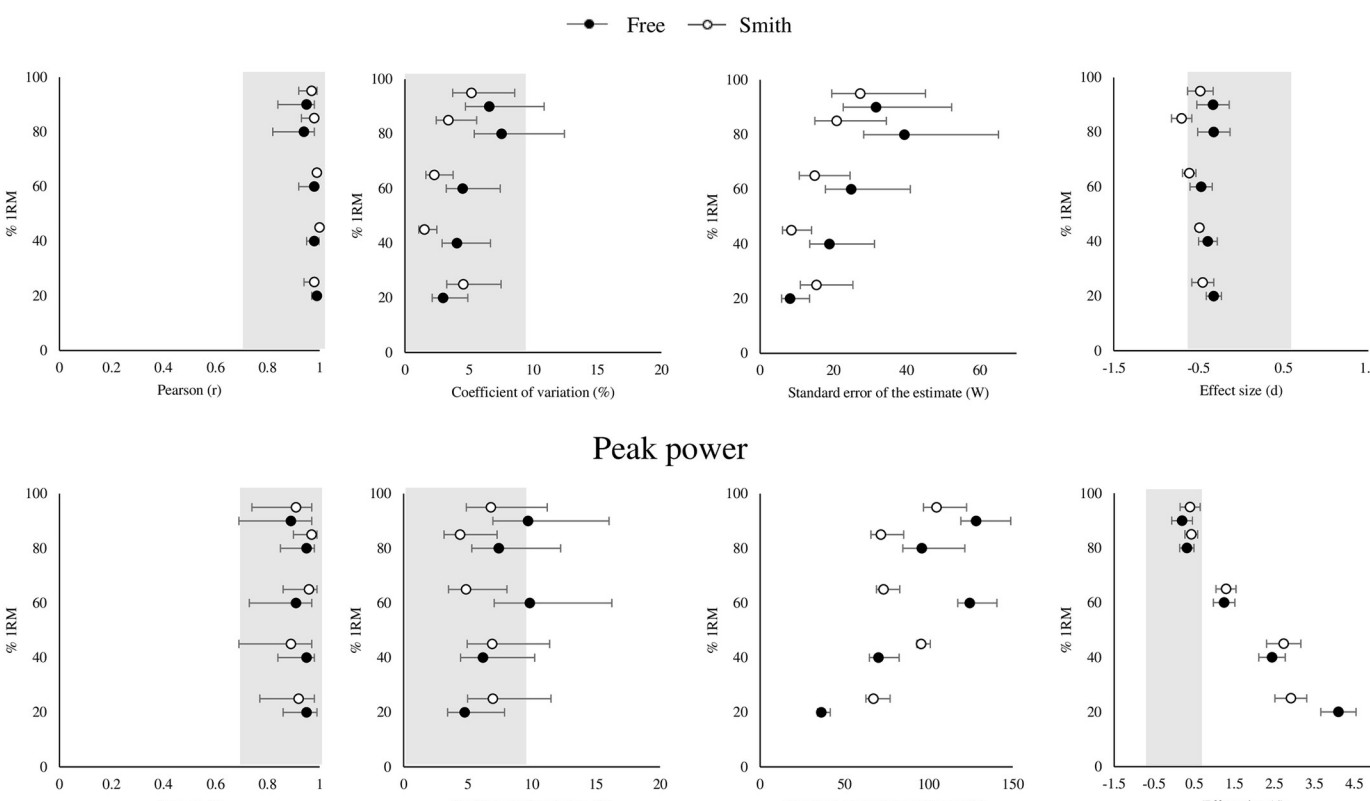

**Fig 2. Validity of Vitruve for the measurement of the mean and peak power at different relative loads during the free-weight and Smith machine back squat exercises.** Forest plots displaying Pearson correlation coefficient (r), coefficient of variation (%), standard error of the estimate (W) and effect size (d). In grey the threshold for validity.

For PP, Vitruve did not meet the validity criteria for any of the relative loads during the free-weight back squat exercise mode (20 to 60% 1RM: ES $\geq$ 1.3; 40 to 90% 1RM: CV 95% CI > 10%). During the Smith machine back squat exercise mode, Vitruve did not meet the validity criteria for any of the relative loads except for the 80% 1RM ($r$ = 0.97; CV = 4.4%; ES = 0.43). A positive fixed bias was observed during the free-weight (from 20% 1RM to 80% 1RM: 475 to 96 W) as well as during the Smith machine (from 20% 1RM to 90% 1RM: 492 to 96 W) back squat exercise modes. A proportional bias was observed for the lower relative loads during the free-weight ($\leq$ 40% 1RM) and Smith machine ($\leq$ 60% 1RM) back squat exercises ($R^2 \geq$ 0.14).

### Between-day reliability

The SDDs of MV, PV, MP and PP for each relative load and exercise mode are reported in Table 2.

All the variables analyzed met the criterion for acceptable reliability (i.e., CV $\leq$ 10%) (Fig 3). MV and PV displayed the lowest CV ($\leq$ 4.0%) followed by MP (5.7%; MV/PV *vs.* MP: $CV_{ratio} \geq$ 1.42) and PP (7.8%; MV/PV vs. PP: $CV_{ratio} \geq$ 1.36). These variables generally showed greater reliability in the Smith machine exercise mode ($CV_{ratio}$ range: 1.12 to 2.06), and similar reliability to the ones reported by GymAware ($CV_{ratio}$ range: 1.00 to 1.15).

**Table 2. Recommendations for the smallest detectable difference of mean velocity (MV), peak velocity (PV), mean power (MP) and peak power (PP) recorded by the Vitruve device at different relative loads during the free-weight and Smith machine back squat exercise modes.**

| %1RM | MV (m/s) | | PV (m/s) | | MP (W) | | PP (W) | |
|---|---|---|---|---|---|---|---|---|
| | Free-weight | Smith machine | Free-weight | Smith machine | Free-weight | Smith machine | Free-weight | Smith machine |
| 20 | 0.08 | 0.05 | 0.17 | 0.14 | 38 | 23 | 190 | 160 |
| 40 | 0.05 | 0.08 | 0.11 | 0.17 | 103 | 68 | 371 | 566 |
| 60 | 0.03 | 0.03 | 0.14 | 0.11 | 101 | 43 | 355 | 173 |
| 80 | 0.08 | 0.05 | 0.14 | 0.11 | 109 | 53 | 291 | 308 |
| 90 | 0.08 | 0.05 | 0.14 | 0.14 | 134 | 84 | 339 | 411 |

%1RM: percentage of the one-repetition maximum

## Discussion

This study was designed to examine the concurrent validity of the new Vitruve linear position transducer for measuring MV, PV, MP, and PP during the free-weight and Smith machine back squat exercises at different relative loads compared to GymAware linear position transducer. The between-day reliability of these variables displayed by Vitruve was also examined. Considering GymAware as a reference point, Vitruve was not valid for measuring velocity and power outcomes. Only an acceptable validity was observed for the PV in the Smith machine back squat, being in the rest of the variables, relative loads, and exercise mode mostly inaccurate. The MV and PV reported by Vitruve displayed superior reliability compared to the MP and PP. Similarly, all variables were more reliable in the Smith machine compared to the free-weight back squat exercise mode.

Vitruve underestimated the MV reported by GymAware at all relative loads during both back squat exercise modes. These results are in line with the ones reported by Külkamp et al. [8], where the bar displacement reported by Vitruve overestimated the actual heights established on a Smith machine by 1–2 cm on average, which would then cause a lower MV for a given lift time. Additionally, the findings of the present study agree with those from Kilgallon et al. [9] who reported an underestimation of the actual free-weight back squat 1RM (13.5 [-47.8 to 20.6] kg) estimated through the load-velocity profile (20 to 90% 1RM) using the MV. In contrast to the findings for MV displayed by Vitruve in the present study, PV met the validity criteria for all relative loads in the Smith machine back squat exercise, and in the free-weight back squat exercise except for the 20 and 60% 1RM loads. Interestingly, Kigallon et al. [9] observed that the free-weight back squat 1RM estimations were overall more accurate using the PV, although large random errors were found (2.8 [-29.4 to 35.0] kg). Several factors could be behind these underestimations of MV, and the higher validity observed for PV. For example, GymAware, unlike Vitruve, includes an angle sensor that corrects the vertical displacement according to the horizontal motion of the lift [11]. Additionally, the data processing performed by the software of each device might cause a different start and end point selection of the concentric phase. This has previously been described to be the main cause for discrepancies between mean and peak variables such as force, velocity, and power [26–28].

Regarding the power variables, Vitruve only met the validity criteria for MP with 20 and 40% 1RM in both back squat exercise modes, and none of the conditions met the validity criteria for PP. The discrepancy between the devices might be due to filtering and smoothing techniques applied to the raw data, as well as the double differentiation of displacement data to derive acceleration, from which power measures are obtained [11, 29]. In this regard, Vitruve samples data at 100 Hz [20], which is within the recommended sampling frequency range for

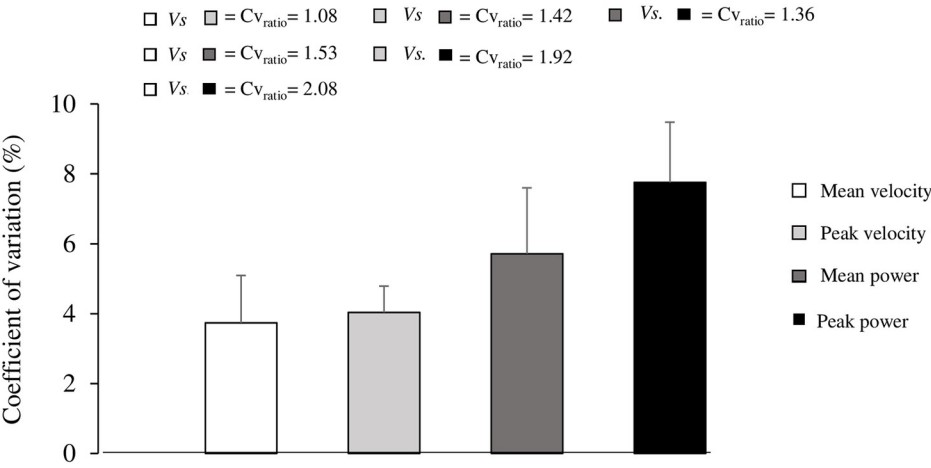

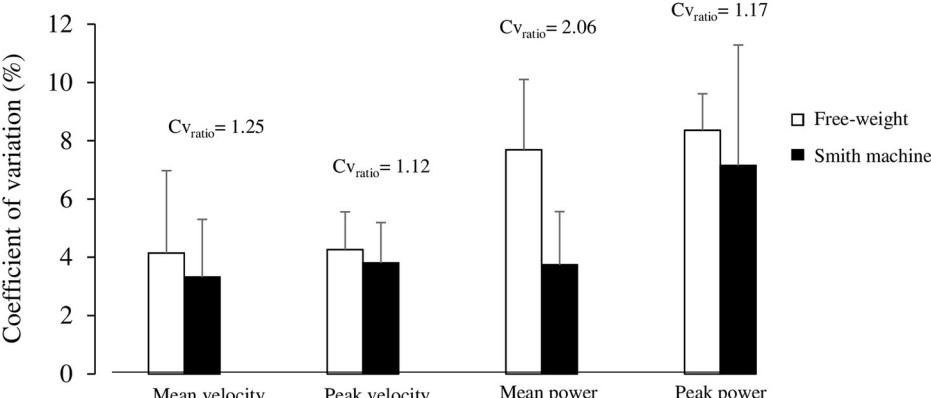

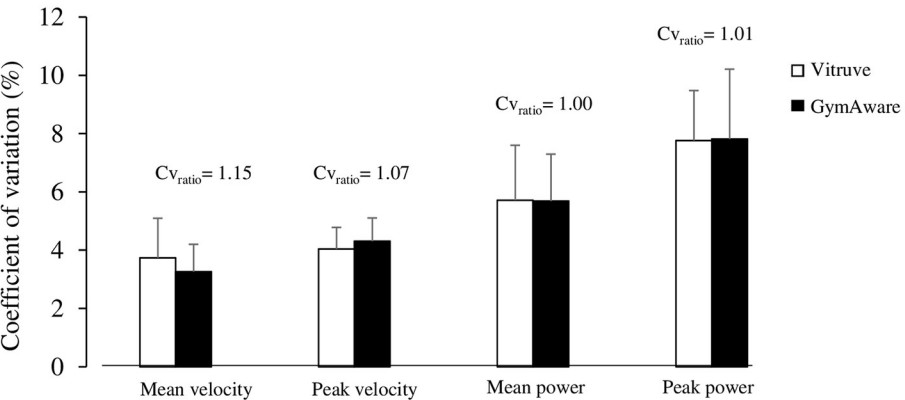

**Fig 3. Absolute reliability compared between variables, exercise modes and devices.** $CV_{ratio}$, Coefficient of variation ratio.

the squat exercise (i.e., 100 to 200 Hz) [11, 29]. In contrast to Vitruve and continuous position counting of traditional sampling methods, GymAware adopts a variable sampling rate through which the sensor only records the cable position when there is a transition [19]. This removes the noise associated with the periodic sampling [16] and thus may have caused discrepancies between the measurements obtained by Vitruve and GymAware.

A similar between-day reliability was observed for the velocity and power variables reported by Vitruve and GymAware. To correctly interpret the results observed, it is crucial to differentiate those studies analyzing the biological error (i.e., variability) in conjunction with the technological error, and those that isolate the technological error using two units of the same linear position transducer [30]. In this regard, Jukic et al. [31] have reported the technological error of GymAware for the MV and PV during the free-weight back squat exercise using two units positioned at each side of the barbell, establishing a 0.05 and 0.10 m/s threshold for detecting a meaningful performance change through this device, respectively. Considering that the biological error could not be removed in the present study, larger SDD values were found across different relative loads (20 to 90% 1RM) for the MV (from 0.03 to 0.08 m/s) and PV (from 0.11 to 0.17 m/s) using the Vitruve linear position transducer. Interestingly, Kilgallon et al. [9] reported different SDD for MV (from 0.08 to 0.10 m/s) and PV (from 0.05 to 0.10 m/s) recorded by Vitruve. However, it should be noted that Kilgallon et al. [9] recruited considerably stronger participants in their study (free-weight back squat 1RM: 132 kg vs. 178 kg) suggesting that SDD values for kinematic variables could be population specific. Indeed, similar SDD values for MV (from 0.06 to 0.08 m/s) and PV (from 0.11 to 0.19 m/s) were previously reported for the participants of similar characteristics to the ones in the present study (1RM free-weight back squat: 132 *vs*. 142 kg) using a different linear position transducer (PT5A-250; Celesco) [24]. Therefore, practitioners should consider these SDD values when interpreting meaningful changes in performance while also having in mind the population they are working with. In this regard, it is also worth remembering that the horizontal motion of the lift could compromise these SDD values, for which practitioners should ensure a correct lift execution [10]. Likewise, the squat depth and the data processing performed by the software of each device should be considered since a different start point of the concentric phase could compromise the mean and peak values recorded by the device.

Lastly, a few limitations should be acknowledged. The absence of data for certain relative loads (30%, 50%, and 70% 1RM) may impact the comprehensiveness of our analyses. Although the validity of GymAware has been established in several studies (12–14), Vitruve was not compared to a gold-standard device, such as a 3D motion capture system. Lastly, while we included power variables in our analyses, it is important to recognize the lack of comparison with direct force measurements. Future research lines should address these knowledge gaps, as well as the analysis of the validity and reliability of Vitruve in other exercise modes (e.g., bench press, deadlift).

## Conclusions

Considering GymAware as a reference point, the Vitruve system demonstrated insufficient validity for measuring velocity and power outcomes. Acceptable validity was observed only for PV in the Smith machine back squat, while relative loads and other exercise modes yielded inaccurate results. Conversely, users of the Vitruve system should note that MV and PV exhibited greater reliability compared to MP and PP. Additionally, all variables measured by Vitruve showed higher reliability in the Smith machine than in the free-weight back squat exercise mode. If practitioners choose to use the Vitruve device despite its poor validity relative to GymAware, they are encouraged to pay special attention to the SDD reported in this study for

the velocity and power metrics obtained in both the free-weight and Smith machine back squat exercises. This consideration is essential for accurately interpreting meaningful changes in performance.

## Supporting information

**S1 Checklist. Inclusivity in global research.**
(DOCX)

**S1 Database.**
(XLSX)

## Acknowledgments

The authors would like to thank all participants.

## Author Contributions

**Conceptualization:** Deniz Şentürk, Zeki Akyildiz, Onat Çetin, Selman Kaya, Alejandro Pérez-Castilla, Ivan Jukic.

**Data curation:** Deniz Şentürk, Zeki Akyildiz, Onat Çetin, Selman Kaya, Alejandro Pérez-Castilla, Ivan Jukic.

**Formal analysis:** Santiago A. Ruiz-Alias, Alejandro Pérez-Castilla.

**Investigation:** Deniz Şentürk, Zeki Akyildiz, Onat Çetin, Selman Kaya.

**Methodology:** Ivan Jukic.

**Project administration:** Deniz Şentürk, Zeki Akyildiz, Ivan Jukic.

**Resources:** Deniz Şentürk.

**Software:** Santiago A. Ruiz-Alias.

**Supervision:** Ivan Jukic.

**Validation:** Ivan Jukic.

**Visualization:** Santiago A. Ruiz-Alias.

**Writing – original draft:** Santiago A. Ruiz-Alias, Deniz Şentürk, Zeki Akyildiz, Onat Çetin, Selman Kaya.

**Writing – review & editing:** Santiago A. Ruiz-Alias, Zeki Akyildiz, Alejandro Pérez-Castilla, Ivan Jukic.

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
