## [Decision Letter · Decision Letter 0]

24 Jun 2024

PONE-D-24-16910Validity and Reliability of Velocity and Power Measures Provided by the Vitruve Linear Position Transducer

PLOS ONE

Dear Dr. Jukic,

Thank you for submitting your manuscript to PLOS ONE. After careful consideration, we feel that it has merit but does not fully meet PLOS ONE’s publication criteria as it currently stands. Therefore, we invite you to submit a revised version of the manuscript that addresses the points raised during the review process.

Dear Authors,

Two independent experts have reviewed your manuscript. They find it interesting and rigorous but agree that your conclusions need to be stronger and more straightforward regarding the validity and practical usefulness of the Vitruve Linear Position Transducer. Since the device did not achieve high levels of validity, the reviewers suggest rewriting the parts of the manuscript.

We look forward to receiving your revised manuscript.

Kind regards,

Danica Janicijevic, Ph.D

Academic Editor

PLOS ONE

Journal Requirements:

Reviewers' comments:

Reviewer's Responses to Questions

**Comments to the Author**

1. Is the manuscript technically sound, and do the data support the conclusions?

Reviewer #1: Yes

Reviewer #2: Partly

2. Has the statistical analysis been performed appropriately and rigorously? 

Reviewer #1: Yes

Reviewer #2: No

3. Have the authors made all data underlying the findings in their manuscript fully available?

Reviewer #1: Yes

Reviewer #2: Yes

4. Is the manuscript presented in an intelligible fashion and written in standard English?

Reviewer #1: Yes

Reviewer #2: Yes

5. Review Comments to the Author

Reviewer #1: The Vitruve clearly didn't measure what it said it would measure. I understand you likely don't want to make too strong of statements in the paper, but the data wasn't valid for MV, PP, MP, and PV. We also can see that it more or less only worked ok with a Smith machine, which few people use in real life. Considering all of this, I wouldn't focus on the fact that the device cannot be used interchangeably with other devices, but I would simply advise against using it. It doesn't work good enough for academic researchers to recommend its use "even if you only use it and no other equipment". If readers don't know that you can't use data from different devices by now, they wouldn't learn anything new by stating that in this paper. Thus, please rework the discussion and conclusions to indicate that the device should not be used (not even "if you accept X amount of error"). In summary, I would like to see stronger statements. If you bought this device and used it, would you be entirely happy with the data? If so, no need to have stronger statements. However, if you wouldn't be pleased with the device, other people wouldn't either. Therefore, the story goes beyond the data.

Reviewer #2: GENERAL COMMENT

The study determined the validity and between-day reliability of the mean velocity, peak velocity, mean power, and peak power provided by the Vitruve device at different relative loads in the free-weight and Smith machine back squat. In general, the results showed that the Vitruve did not meet the validity criteria for almost all variables analyzed but presented acceptable reliability for all variables analyzed. I congratulate the authors for their time and effort in conducting this study. The experimental procedures seem rigorous, but the introduction, discussion, statistical analysis, and reporting of results can be improved. The reasons for conducting the study and its pertinence should also be reinforced (as I understood, at least two studies have already explored this theme). Furthermore, the small sample size of fourteen male sports science students should be adequately justified in the methods and addressed in a limitations section. Finally, although the Viturve seems reliable, it is not valid, especially when measuring the mean velocity. Therefore, I cannot entirely agree with the authors’ statement that Vitruve seems useful for resistance training, monitoring, and prescription. This critical information should be well revised throughout the manuscript. Major revisions should be made. Below, I provide some comments and suggestions.

Best regards

ABSTRACT

LL57-58: Please rewrite the sentence for clarity.

LL63-66: According to your data, Vitruve is reliable but does not present valid measures of velocity and power across a broad range of relative loads. In other words, Vitruve is consistent but does not accurately measure what it is meant to measure, especially the mean velocity (one of the most reported variables in velocity-based RT interventions). Therefore, if a device is not valid, it is useless. Please consider rewriting the sentence.

INTRODUCTION

- LL72-77: Please consider citing the review conducted by Badillo et al. (2022), as it addresses the core concepts of velocity-based resistance training (10.1186/s40798-022-00513-z).

- LL78: “Different velocity-monitoring systems are available nowadays.” Please consider citing some studies.

- LL79: “…valid and reliable devices providing barbell velocity data are needed”. Please consider adding a brief definition of validity and reliability.

- LL84-86: I would complement this sentence by adding linear velocity transducers.

- LL88-90: Please consider citing the review conducted by Moreno-Villanueva on this topic (10.1080/14763141.2021.1988136).

- LL96-107: These lines are important as they formulate the research problem. However, what is the main difference between the current study and those previously conducted? This information should be much more explicit so that readers can understand the current study’s need and novelty.

- LL108-110: Please add references.

- LL115: Why the GymAware and not a different device? The reason(s) for introducing this device in the narrative must be well-defined.

- LL122-124: What is the need/relevance to include power variables? Are the estimates of power accurate in these devices? Previous authors have alerted for inaccurate power measures when using barbell kinematics (10.1519/JSC.0b013e31822e7b48). Considering that a force plate must be used for accurate power estimates, I question the utility of reporting power variables in the study.

- LL130: Please consider formulating the study’s hypotheses based on previous research.

MATERIALS AND METHODS

- LL152: “Fourteen male sports science students…”. I recommend the authors present a reasonable sample size estimation based on an expected reliability value. Please also indicate the statistical power of the study.

- LL162-187: Please add references for these procedures.

- LL188-195: Please add references for these procedures.

- LL198-202: Was the absolute load adjusted so that the velocity achieved corresponded to the relative intensity? How can the authors guarantee that the velocities reached during sessions correspond to the actual velocities of the participants? This aspect should be clear, considering the typical fluctuations of 1RM from day to day, thus affecting all relative loads.

- LL229-231: The CV (%) is more commonly associated with reliability, as it indicates the degree of variation or dispersion between measures (i.e., the consistency of a measurement device).

- In the statistical analysis subsection, I recommend that the authors calculate and report the intraclass correlation coefficient (ICC), Lin’s concordance correlation coefficient (CCC), and linear regression (r2 and SEE) to increase the robustness of their analysis. These tests have been consistently used in similar research; thus, they should be reported for further comparisons.

- LL242-245: Please briefly describe how the CV ratio was calculated.

RESULTS

This section must be updated to integrate the results of the tests recommended before (ICC, CCC, and regression analysis). Furthermore, I recommend presenting the Bland-Altman results in a figure, not a table, since visual inspection, in this case, is more direct in interpreting results. Regression lines with the respective r2 and SEE values must also be shown to analyze the concurrent validity between devices.

DISCUSSION

- LL327: I recommend replacing the word “monitoring” with “measuring”.

- LL330-331: “The results revealed that Vitruve and GymAware cannot be used interchangeably.” If we consider the GymAware device as the gold standard, this sentence means that the Vitruve is not a valid device for measuring velocity and power. I suggest rewriting the sentence for clarity.

- LL342-344: According to these lines, only the measurement of PV during the Smith machine back squat seems valid. Therefore, the practical implications paragraph and the conclusions sentences should adequately address these results to clarify the readers. Vitruve is not valid for measuring MV during the back squat in both variants, but it is valid for measuring PV only during the Smith machine back squat.

- LL347-348: “Several factors could be behind these underestimations of MV, and the higher validity observed for PV”. “For MV, Vitruve did not meet the validity criteria for any of the relative loads during the free-weight back squat exercise mode or Smith machine back squat exercise mode». The sentence must be reformulated to clearly highlight this information.

- LL349: It is said that the “GymAware … includes an angle sensor that corrects the vertical displacement according to the horizontal motion of the lift”. However, what is the influence of this sensor? Is it really that relevant?

- LL350: “Additionally, the data processing performed by the software of each device might cause a different start and end point selection of the concentric phase.” How does the mechanism differ? In lines 213-214, the authors indicate, “Both linear position transducers internally collect and process the data in a similar fashion”. Please reformulate for clarity.

- I recommend that the authors address some of the study’s limitations. For example, some limitations include the low sample size, the absence of velocity data for some relative loads (30, 50, and 70% 1RM), and the lack of a gold standard device such as a 3D motion capture system to measure velocity. Furthermore, if power variables are maintained in the analysis, the lack of a direct force measurement to determine power must be recognized.

- I also recommend that the authors provide future research lines on this topic and practical recommendations for coaches and researchers regarding using the Vitruve device in the free-weight and Smith machine back squat. What should and should not be used must be mentioned.

6. PLOS authors have the option to publish the peer review history of their article (what does this mean?). If published, this will include your full peer review and any attached files.

Reviewer #1: No

Reviewer #2: No

---

## [Author Response · Author response to Decision Letter 0]

20 Aug 2024

Please see the Word document attached to this submission.

---

## [Decision Letter · Decision Letter 1]

2 Sep 2024

PONE-D-24-16910R1Validity and Reliability of Velocity and Power Measures Provided by the Vitruve Linear Position TransducerPLOS ONE

Dear Dr. Jukic,

Thank you for submitting your manuscript to PLOS ONE. After careful consideration, we feel that it has merit but does not fully meet PLOS ONE’s publication criteria as it currently stands. Therefore, we invite you to submit a revised version of the manuscript that addresses the points raised during the review process.

Dear authors, one of the two reviewers has provided one more general comment on the article that I would like to see addressed. 

We look forward to receiving your revised manuscript.

Kind regards,

Danica Janicijevic, Ph.D

Academic Editor

PLOS ONE

Journal Requirements:

Reviewers' comments:

Reviewer's Responses to Questions

**Comments to the Author**

1. If the authors have adequately addressed your comments raised in a previous round of review and you feel that this manuscript is now acceptable for publication, you may indicate that here to bypass the “Comments to the Author” section, enter your conflict of interest statement in the “Confidential to Editor” section, and submit your "Accept" recommendation.

Reviewer #1: (No Response)

Reviewer #2: All comments have been addressed

2. Is the manuscript technically sound, and do the data support the conclusions?

Reviewer #1: Yes

Reviewer #2: Yes

3. Has the statistical analysis been performed appropriately and rigorously? 

Reviewer #1: Yes

Reviewer #2: Yes

4. Have the authors made all data underlying the findings in their manuscript fully available?

Reviewer #1: Yes

Reviewer #2: Yes

5. Is the manuscript presented in an intelligible fashion and written in standard English?

Reviewer #1: Yes

Reviewer #2: Yes

6. Review Comments to the Author

Reviewer #1: Thank you, some of the claims are better. However, when using words like "superior" it makes it sound like something is actually good. If you eat pizza and the taste of one pizza is 4/10 and the other is 5/10, the second is superior than the other. Despite being superior, would you want to eat that pizza? No. I feel the same with this paper... we have tons of devices to choose from, some of which are excellent at different loads, different exercises, with machines, with free weights etc. Yet, here we have a device that isn't as good (according to the data presented). So yea, although the data show the Vitruve is better at measuring V than P, that it's better in a machine than with a free weight, it seems to be a poor choice for people who will actually use this. Remember, even if you hedge your statements a bit, authors will read your title, abstract, and conclusion and still think this device is good (it's just not excellent at everything). However, compared to the other options out there, this device doesn't seem to be up to par yet.

In no way am I involved with any company, I don't want to promote anything, and I don't want to convince you (or potential readers) what devices they should or shouldn't use. However, when looking at this data, and when looking at other similar data on other devices, these data indicate that the device shouldn't be used in its current state. If you agree, please make some more changes throughout the document, especially at strategic points. However, if you have a different opinion and you feel as though you want to highlight the positive aspects of the data (compared to the overall negative aspects), I cannot stop you because it's just the way you choose to report and interpret the data. There would not be anything scientifically wrong, per se.

Reviewer #2: Dear authors,

Congratulations on your excellent work revising the manuscript as requested and answering all my comments and suggestions. In my opinion, the manuscript is ready for publication.

Best regards

7. PLOS authors have the option to publish the peer review history of their article (what does this mean?). If published, this will include your full peer review and any attached files.

Reviewer #1: No

Reviewer #2: No

---

## [Author Response · Author response to Decision Letter 1]

3 Oct 2024

Reviewer #1: 

COMMENT:

Thank you, some of the claims are better. However, when using words like "superior" it makes it sound like something is actually good. If you eat pizza and the taste of one pizza is 4/10 and the other is 5/10, the second is superior than the other. Despite being superior, would you want to eat that pizza? No. I feel the same with this paper... we have tons of devices to choose from, some of which are excellent at different loads, different exercises, with machines, with free weights etc. Yet, here we have a device that isn't as good (according to the data presented). So yea, although the data show the Vitruve is better at measuring V than P, that it's better in a machine than with a free weight, it seems to be a poor choice for people who will actually use this. Remember, even if you hedge your statements a bit, authors will read your title, abstract, and conclusion and still think this device is good (it's just not excellent at everything). However, compared to the other options out there, this device doesn't seem to be up to par yet. In no way am I involved with any company, I don't want to promote anything, and I don't want to convince you (or potential readers) what devices they should or shouldn't use. However, when looking at this data, and when looking at other similar data on other devices, these data indicate that the device shouldn't be used in its current state. If you agree, please make some more changes throughout the document, especially at strategic points. However, if you have a different opinion and you feel as though you want to highlight the positive aspects of the data (compared to the overall negative aspects), I cannot stop you because it's just the way you choose to report and interpret the data. There would not be anything scientifically wrong, per se.

RESPONSE:

First of all, the authors would like to thank the reviewer for their constructive comments. The comments have significantly improved the quality of the manuscript and we are grateful for that.

Regarding the suggestion to adopt a more conservative tone in certain sections of the manuscript, we would like to address this concern. The term "superior" is specifically used when comparing exercise modes (Smith vs. Free), and we believe that readers will understand this distinction is not a claim of good performance relative to our reference system. Nevertheless, we have taken the reviewer's perspective into account and rewritten the conclusion as follows:

"Considering GymAware as a reference point, the Vitruve system demonstrated insufficient validity for measuring velocity and power outcomes. Acceptable validity was observed only for PV in the Smith machine back squat, while relative loads and other exercise modes yielded inaccurate results. Conversely, users of the Vitruve system should note that MV and PV exhibited greater reliability compared to MP and PP. Additionally, all variables measured by Vitruve showed higher reliability in the Smith machine than in the free-weight back squat exercise mode. If practitioners choose to use the Vitruve device despite its poor validity relative to GymAware, they are encouraged to pay special attention to the SDD reported in this study for the velocity and power metrics obtained in both the free-weight and Smith machine back squat exercises. This consideration is essential for accurately interpreting meaningful changes in performance."

In this manner, we aim to clarify that we, the authors, are not recommending the use of Vitruve; rather, we present statements based on the results of its validity and reliability analysis. We hope we have made this clearer now, and again, we really appreciate your comments on this.

We also amended the “conclusion” section of the abstract as follows: "Acceptable validity was observed only for PV in the Smith machine back squat, while the other variables—regardless of relative loads and exercise modes—were mostly inaccurate. All variables demonstrated acceptable reliability, with greater reliability noted in the Smith machine compared to the free-weight back squat exercise mode."

We really hope that the changes made address your comments.

---

## [Editor Report · Decision Letter 2]

7 Oct 2024

Validity and Reliability of Velocity and Power Measures Provided by the Vitruve Linear Position Transducer

PONE-D-24-16910R2

Dear Dr. Ivan Jukic,

We’re pleased to inform you that your manuscript has been judged scientifically suitable for publication and will be formally accepted for publication once it meets all outstanding technical requirements.

Kind regards,

Danica Janicijevic, Ph.D

Academic Editor

PLOS ONE

Additional Editor Comments (optional):

Congratulations!
---

## [Editor Report · Acceptance letter]

15 Oct 2024

PONE-D-24-16910R2 

PLOS ONE

Dear Dr. Jukic, 

I'm pleased to inform you that your manuscript has been deemed suitable for publication in PLOS ONE. Congratulations! Your manuscript is now being handed over to our production team.

Kind regards, 

on behalf of

Dr. Danica Janicijevic 

Academic Editor

PLOS ONE